Physical fitness predicts technical-tactical and time-motion profile in simulated Judo and Brazilian Jiu-Jitsu matches

Coswig Victor S. vcoswig@ufpa.br vcoswig@gmail.com 1
Gentil Paulo 2
Bueno João C.A. 3
Follmer Bruno 4
Marques Vitor A. 2
Del Vecchio Fabrício B. 5
1 Faculdade de Educação Física, Universidade Federal do Pará , Castanhal , Pará , Brasil
2 Faculdade de Educação Física e Dança, Universidade Federal de Goiás , Goiania , Brasil
3 Faculdade de Educação Física, Universidade Federal do Paraná , Curitiba , Brazil
4 Laboratório de Biomecânica, Universidade Federal de Santa Catarina , Florianópolis , Brasil
5 Escola Superior de Educação Física, Universidade Federal de Pelotas , Pelotas , Brasil
Ramírez-Campillo Rodrigo
Electronic publication date: 2018 May 25
Publication date: 2018
Volume: 6
Electronic Location ID: e4851
Received 2018 Mar 8; Accepted 2018 May 6
Copyright: ©2018 Coswig et al.
Copyright year: 2018
Copyright holder: Coswig et al.
License: This is an open access article distributed under the terms of the Creative Commons Attribution License, which permits unrestricted use, distribution, reproduction and adaptation in any medium and for any purpose provided that it is properly attributed. For attribution, the original author(s), title, publication source (PeerJ) and either DOI or URL of the article must be cited.
License URL: https://creativecommons.org/licenses/by/4.0/

Keywords: Physical effort, Martial arts, Athletic performance, Physical education and training, Software validation

Funding: The authors received no funding for this work.

==============================
Background

Among combat sports, Judo and Brazilian Jiu-Jitsu (BJJ) present elevated physical fitness demands from the high-intensity intermittent efforts. However, information regarding how metabolic and neuromuscular physical fitness is associated with technical-tactical performance in Judo and BJJ fights is not available. This study aimed to relate indicators of physical fitness with combat performance variables in Judo and BJJ.

Methods

The sample consisted of Judo (n = 16) and BJJ (n = 24) male athletes. At the first meeting, the physical tests were applied and, in the second, simulated fights were performed for later notational analysis.

Results

The main findings indicate: (i) high reproducibility of the proposed instrument and protocol used for notational analysis in a mobile device; (ii) differences in the technical-tactical and time-motion patterns between modalities; (iii) performance-related variables are different in Judo and BJJ; and (iv) regression models based on metabolic fitness variables may account for up to 53% of the variances in technical-tactical and/or time-motion variables in Judo and up to 31% in BJJ, whereas neuromuscular fitness models can reach values up to 44 and 73% of prediction in Judo and BJJ, respectively. When all components are combined, they can explain up to 90% of high intensity actions in Judo.

Discussion

In conclusion, performance prediction models in simulated combat indicate that anaerobic, aerobic and neuromuscular fitness variables contribute to explain time-motion variables associated with high intensity and technical-tactical variables in Judo and BJJ fights.

Introduction

Judo and Brazilian Jiu-Jitsu (BJJ) are both grappling Combat Sports, takedowns are the main technical actions in the first and submissions and immobilizations are more frequent in the second (Amtmann & Cotton, 2005; Andreato et al., 2016). Despite the expressive growing in physiological, physical and technical-tactical studies in Combat Sports, associations among those variables was only partially investigated in Olympic Wrestling (Cvetković et al., 2005) and Karate (Blazević et al., 2006; Roschel et al., 2009). Considering high popularity and relevance of Judo and BJJ, the absence of specific information about these modalities represents an important gap in Combat Sports Science.

Combat Sports are characterized by high physical demands resulting from the acyclic patterns and from the high intensity intermittent efforts followed by brief periods of partial recovery (Amtmann & Cotton, 2005; James, 2014). Therefore, essential physical fitness variables are associated with competitive success of these modalities and can be divided into metabolic components, such as aerobic power and anaerobic capacity (Amtmann & Cotton, 2005; James, 2014), and neuromuscular components, such as handgrip endurance, flexibility and muscle strength and power (Amtmann & Cotton, 2005; James, 2014).

Physical training process can become even more specific and efficient when considering the pattern of motor gestures from the technical-tactical analysis (Miarka et al., 2011). Furthermore, knowledge about the fighting pattern contributes to enhance fighters capability to make decisions based on the understanding of their behavior and of their opponents actions (Sterkowicz-Przybycień, Miarka & Fukuda, 2017). The technical-tactical analysis of Judo and BJJ has involved the quantification of the main techniques that generate points, the total frequency of actions, the effectiveness of actions, as well as the directions of projection and types of grasp in the Gi, which is the traditional uniform for belt and jacket grappling modalities (Franchini, Artioli & Brito, 2013).

Despite the high social representation that Judo and BJJ worldwide, little is known about the relationship between physical fitness variables and technical-tactical aspects in grappling modalities. In wrestling athletes, medicine ball throws, 20-m runs and push-ups tests seems to predict (36 to 41%) the variance of three of the five techniques studied (Cvetković et al., 2005). With Judo athletes, Detanico et al. (2012) aimed to relate aerobic and neuromuscular indices with Judo specific actions; however, relations with technical-tactical aspects in combats were not investigated.

Whilst previous studies showed associations between physical and technical aspects in many combat sports, investigations have been conducted based on isolated technical actions which leaves two main points to be investigated: (i) the relationship between physical fitness variables and technical-tactical actions during real or simulated combats and; (ii) responses of these variables in belt and jacket grappling combat sports as Judo and BJJ. The understanding of the relationship between physical fitness and technical-tactical variables can be of great practical importance to coaches and athletes. In this context, the present investigation aimed at studying the relationship between physical fitness variables and technical-tactical analysis in Judo and BJJ athletes .

Methods

Experimental approach to the problem

Athletes visited the laboratory in two opportunities, on the first they signed the informed consent term and performed anthropometric and physical fitness tests. Physical fitness was evaluated in this order: anaerobic capacity, flexibility, vertical and horizontal jumps, judogi dynamic and isometric chin-up tests, medicine ball throws, upper limbs muscular and abdominal endurance and aerobic power. These variables and tests were chosen based on modalities demands and in order to increase external validity and strengthen its practical applications. The second visit involved simulated matches and occurred two to seven days after the first. These matches were recorded in video for further time-motion and technical-tactical analysis, which were considered as dependent variables. The study was approved by the Federal University of Goiás research ethics committee (no 1.679.225/2016) and was performed in accordance with the Declaration of Helsinki ethical standards.

Subjects

The sample was composed of experienced Judo (n = 16) and BJJ (n = 24) male athletes. Judo athletes were 22 ± 6 years, 77.5 ± 11 kg of body mass, 169.3 ± 8 cm of height and 173.5 ± 10 cm of wingspan, while BJJ athletes were 24 ± 4 years, 81.2 ± 9 kg of body mass, 177 ± 7 cm of height and 180 ± 8 cm of wingspan.

To be included in the study, volunteers should meet the following criteria: be engaged in uninterrupted training in the modality during the last three months, practice at least three times a week, be graduated as purple, brown or black belt, belong to adult category (18 to 30 years), have declared to achieve regional competitive level or higher and have participated in an official competition, at least once in the last year. Subjects were excluded when presented locomotor injury in the last 30 days, exhibited functional limitations due to previous injury, performing rapid weight loss practices and did not complete, for any reason, the battery of physical tests or did not finish the simulated combat.

Procedures

Athletes were recruited by convenience through verbal invitation from four different training centers, each one in a different State in Brazil, from south and center-west regions, which characterized this as a multicenter project. In the same way, training centers were chosen by convenience considering previous institutional links, as well personal and material availability.

Pre-tests Warm-up

After body mass and height measurements, subjects underwent a general standardized 10-min warm-up composed by: (i) moderate intensity run (four min); (ii) Jumping jacks and vertical jumps (2 min); (iii) Joint movements and mobility (2 min), and; (iv) tests’ specific movements (2 min). Then, physical fitness tests were applied with a minimum of 15 min between the first (anaerobic capacity) and the second test (Flexibility). The additional tests were separated by 5 min rest periods, except between the judogi chin-up tests, where a 15 min pause is recommended (Branco et al., 2017). The tests were conducted as described below and in this order.

Anaerobic capacity (Cunninghan and Faulkner Test): Athletes performed a specific warm up on the treadmill (alternating walking and sprints of 1 min × 1 min for 4 to 6 min). During the test, the treadmill (Embreex 565 TX-1) speed was set at 13 km/h with a slope of 20% (Cunningham & Faulkner, 1969). The timer was triggered when the athlete started the exercise and was stopped when volitional exhaustion was reached. This test has a high correlation with the RAST (r = 0.89) and Wingate (r = 0.82) tests (Thomas, Plowman & Looney, 2002).

Flexibility (Sit and reach test): Participants were asked to sit with legs full extended and soles feet against a box (Instant Pro Sanny®; Vitco China Manufacturing Co., Ltd., Hong Kong). The athlete bowed slowly and leaned forward as far as he could, sliding his fingers along the ruler (Wells & Dillon, 1952). The distance reached was the final score and after three attempts the highest value was considered for analysis.

Lower Limbs Power (Sargent Jump Test): From the standing position, participants were asked to raise their arms and make a mark (impregnated with chalk powder) at the highest possible point without raising his heels. The athlete then performed a rapid knee flexion and a subsequent countermovement jump, without walking or taking distance to jump, trying to reach the highest height possible. At the point of greatest vertical displacement, the athlete made a second chalk mark on the wall with his fingers. The result was recorded by measuring the distance between the first and second marks (Mackenzie, 2005). Three attempts were performed and the highest value was considered for analysis. This procedure has a high validity when compared to jumps platform (r = 0.99, p = 0.001) and high reproducibility both intra (r = 0.99, p = 0.001) and inter (r = 1.0, p = 0.001) evaluators (Salles et al., 2012).

Lower Limbs Power (Horizontal Jump): A metallic non-elastic tape was previously fixed to the ground, perpendicular to the start line. The athlete remained with feet separated and parallel, positioned behind the start line. Then, athletes were asked to perform shoulder flexion by positioning the arms behind the trunk and flexing the knees. The measurement was performed from the starting line to the place where the athlete first touched the ground (Mackenzie, 2005). Three attempts were performed and the highest value was considered in the analyzes.

Specific Upper Limbs Dynamic Endurance (Dynamic Judogi Chin-up test- DJCT): The athlete positioned himself in a vertical suspension, with fully elbow extension, without ground contact, and hands remained fixed on the Gi. From this position, the participant lifted his body until complete an elbow flexion, then returning to the initial position in a controlled pace. Every successful repetition was counted, and test was over due to concentric failure, identified when the athlete was unable to perform fully elbow flexion (Branco et al., 2017).

Specific Upper Limbs Isometric Endurance (Isometric Judogi Chin-up test- IJCT): The participant was asked to sustain their body mass holding the Gi in a fully flexed elbow position, with the chin above the hands. The score was the time that athlete maintained the correct initial isometric position (Branco et al., 2017).

Upper Limbs Power (Medicine Ball Throw): The athlete was asked to sit with knees extended and back fully supported to the wall, while holding a 3 kg medicine ball close to the chest. At the researcher’s signal, the athlete threw the ball as far as possible, keeping his back against the wall. The throw distance was recorded from zero point to the place where the ball touched the ground for the first time (Mackenzie, 2005).

Upper Limbs Endurance (Push-ups test): The athlete positioned himself in a prone position, with his hands on the ground at a distance of 10 to 20 cm from the shoulder line and toes facing forward. During the test, the participant flexed his elbows until approximately 90° and then extended it again (Mackenzie, 2005). The maximum number of correct repetitions in 1-min was recorded.

Abdominal Endurance (Sit-up test): The athlete lied in dorsal decubitus with knees flexed at 90° and arms crossed over the thorax. The researcher fixed the athlete’s feet to the ground. At the signal of the researcher, the athlete initiated the flexing movements of the trunk until touching the knees with the elbows and returned to the starting position (Mackenzie, 2005). The highest number of complete repetitions in 1-min was computed.

Aerobic Power and Speed Related (VO2max/vVO2max): A heart rate monitor (Polar, model F6/Finland) was used to monitor heart rate (HR) after at least 5 min of standing still. The test started at a speed of 8 km/h and was increased by 1 km/h every 2 min (Billat & Koralsztein, 1996), until volitional exhaustion. HRmax was registered immediately pos t-test, and the recovery HR was registered after 1 min. Final velocity reached (vVO2max) was used for further estimation of VO2max (Billat & Koralsztein, 1996), according to the following equation: VO2max = 2,209 + 3,163 × vVO2max + 0,000525542 × vVO2max3.

Simulated Match

Simulated fights occurred in the second meeting, from 48 h to seven days after the first data collection, with the objective of analyzing competitive performance variables associated with the athletes’ fighting pattern. The matches were performed in fightings areas sized between 8 m × 8 m and 10 m × 10 m. Athletes were matched according to body mass, graduation and time of practice. Judo combats lasted 5 min, as recommended by the International Judo Federation (IJF, 2017), while BJJ combats lasted 10 min, according to the competitive standard for black belts determined by the International Brazilian Jiu-Jitsu Federation (IBJJF, 2017). Prior to the fight, a self-selected 10 min warming up was allowed.

All fights followed the rules of the IJF or IBJJF. The only exception was in case of Ippon/submission, which would lead to the end of the fight, but in this situation the combat returned to standing position for continuity until the end of the time, in order to standardize fighting duration. The fights were recorded by two video cameras (front and side) for later technical-tactical and time-motion analysis.

Notational Analysis

For the notational analysis, Windows Media Player software (Microsoft Corporation, Redmond, WA, USA) was used to reproduce the videos. For technical-tactical and time-motion analysis, the Dartfish EasyTag mobile application (EasyTag Note 2.0 10127.0; DartFish, Fribourg, Switzerland), available for Android and iOS, installed on a Motorola branded device (MotoXPlay; Motorola Mobility LLC, Chicago, IL, USA) was used, as previously suggested for this type of analysis (González & Miarka, 2013). The application layout was configured in order to enable the technical-tactical analysis of the two modalities. Ten matches were randomly selected for being re-analyzed in order to test for the reproductibitliy of the method

Technical-tactical analysis

Technical-tactical analysis considered specific motor gestures of the modalities and biomechanical models of analysis based on levers, as originally proposed by Sacripanti (2008). In summary, techniques were categorized as “Physical Lever Throws”, where the body of the opponent turns around a stopping point; or “Couple of Forces Throws”, where opposite forces were applied among upper and lower limbs. This type of analysis is frequently used for the Judo matches (Sterkowicz, Sacripanti & Sterkowicz-Przybycień, 2013; Sterkowicz-Przybycień, Miarka & Fukuda, 2017) and now have been adapted for BJJ. From these, the number of effective actions accounted as the techniques that induce points or submission, while non-effective motor actions were considered the techniques that do not result in scores or submission of the opponent. From these, the effectiveness index was quantified from the equation: Effectiveness=N of effective actions∕N of total actions×100%

Time-motion analysis

The method proposed by Del Vecchio et al. (2007) was used for time-motion analysis, in which the actual combat time (pause time −5 or 10 min), standing fighting time and ground fighting time were quantified. In addition, the intensity of the efforts was subjectively categorized into high and low intensity actions, according to the following characteristics (Andreato et al., 2015b):

i) High intensity: Actions in which the fighter tries to advance, progress or evolve with clear force, muscular strength or power.

ii) Low intensity: Actions with slow movements, with an apparent low level of applied force and power. During the analysis it was considered that the tactical actions to advance are not necessarily carried out in high intensity.

In this sense, the effort:pause (E:P) ratio was calculated, where effort has being considered as the periods between fight and pause command, while pause periods comprised the moments between pause and fight commands, as performed by Del Vecchio et al. (2007). The high:low intensity ratio (HI:LI) was also calculated. All time-motion and technical-tactical analysis were made by a Brazilian Jiu-Jitsu Black Belt, which is also a researcher with high level of specialization in these disciplines.

Statistical Analysis

After Shapiro–Wilk’s test for normality and Levene’s for equality of variances, statistical methods were applied as follows:

Descriptive data are presented by mean ± standard deviation (SD) when normality was confirmed (Physical fitness tests), if not, median, percetiles (25–75%) and coefficient of variation (CV) were used (Time-motion and Technical-tactical analysis).

Spearman coefficient was used to calculate correlations, which were used to define entrance order in the regression analysis models by the Backward method, for which physical fitness variables, categorized as metabolic and neuromuscular, were considered as predictors while technical-tactical and time-motion were considered as outcomes.

For comparisons between modalities, an independent student t test was applied and effect size (ES) was calculated by the following equation: ES=t2∕t2+df, where t is the score of t test and df are the degrees of freedom (Field, 2009). For technical-tactical variables the Mann–Whitney test was applied and ES were calculated by the following equation: ES =Z∕N, where Z corresponds to the Mann–Whitney score and N is the number of observations (Field, 2009). For both, ES was interpreted as small (ES < 0.10), medium (ES ≥ 0.10 and  < 0.50) or large (ES ≥ 0.50) (Field, 2009). To test the mobile application reproducibility the Cohen’s Kappa was applied and the strength of agreement was set as weak (0.0 to 0.2), reasonable (0.21 to 0.40), moderate (0.41 to 0.60), strong (0.61 to 0.80) and almost perfect (0.81 to 1.0) (Field, 2009). All data analyses were executed in IBM SPSS software version 22.0 and statistical significance was set at 5%.

Results

Physical fitness and descriptive analysis

No differences were found between training centers for physical fitness variables, which reinforce sample homogeneity. Anthropometric differences were found for stature (t = 3.09; p = 0.004) and wingspan (t = 2.27; p = 0.02), with higher values for BJJ athletes. Descriptive results of physical fitness tests are presented in Table 1. BJJ athletes showed higher upper limbs power (t = 3.05; p = 0.004) based on medicine ball throws, while Judokas showed higher HRmax after progressive test (t =  − 3.17; p = 0.003).

Table 1 Physical fitness tests descriptive data for Judo (n = 16) and BJJ athletes (n = 24).

	BJJ	JUDO				
	Mean	±SD	CV (%)	Mean	±SD	CV (%)	t	p-value	ES	
C&FAT (s)	66.3	22.9	34.5	58.3	21.1	36.3	1.13	0.26	0.17	
Flexibility (cm)	32.1	7.9	24.6	35.8	7.4	20.7	−1.49	0.14	0.19	
Vertical jump (cm)	46.5	7.7	16.5	44.1	9.5	21.6	0.85	0.39	0.15	
Horizontal jump (cm)	225.9	23.5	10.4	218.3	36.9	16.9	0.80	0.43	0.14	
DJCT (reps)	7.6	3.7	48.7	5.9	4.8	81.2	1.25	0.22	0.18	
IJCT (s)	32.6	12.1	37.2	35.8	23.2	64.8	−0.56	0.58	0.12	
Medicine ball throw (m)	4.3	1.1	26.4	3.3	0.8	23.1	3.05	0.004	0.25	
Push-ups (reps)	36.5	8.8	24.2	36.6	16.6	45.2	−0.02	0.98	0.02	
Sit-ups (reps)	41.7	5.4	13.0	38.1	12.4	32.5	1.26	0.21	0.18	
Rest HR (bpm)	104.8	17.6	16.8	97.0	17.2	17.7	1.38	0.17	0.19	
vVO2max (km/h)	14.5	1.6	11.1	14.3	2.3	15.8	0.49	0.62	0.11	
VO2max (ml/kg/min)	50.9	5.7	11.1	50.1	7.9	15.8	0.39	0.69	0.10	
HRmax (bpm)	185.3	11.6	6.3	194.9	4.6	2.3	−3.17	0.003	0.26	
HR after 1min (bpm)	148.0	13.2	8.9	143.9	20.3	14.1	0.77	0.44	0.14	
Notes.

C&FAT Cunningham and Faulkner Anaerobic Test

DJCT Dynamic Judogui Chin-up test

IJCT Isometric Judogui Chin-up Test

VO2max Maximal Oxygen Uptake

vVO2max Speed related to Maximal Oxygen Uptake

HR Heart Rate

ES Effect size

CV Coefficient of variation

Values in bold indicate medium effect.

Technical-tactical and time-motion analysis

Technical-Tactical data are presented on Table 2 where, depite higher frequency of BJJ actions, effectiveness was higher for Judo athletes. Considering time-motion analysis, the mobile application showed Kappa indices higher than 0.85, which indicates an “almost perfect” strength of agreement for all variables measured. Time-motion comparisons between modalities are presented in Table 3 and Fig. 1. Data in Table 3 indicates that Judokas showed higher relative pause time (U = 2.000; p < 0.001) and higher number of pause blocks (U = 20.500; p < 0.001). BJJ athletes showed higher relative time in high intensity actions (U = 45.500; p < 0.001). Figure 1 presented proportions of E:P ratio and HI:LI ratio by modalities, where Judo values were 3.5:1 and 1:12 and BJJ values were 22:1 and 1:3.5, respectively. When standing and ground positions were considered, the HI:LI ratio on the ground was 1:14 in Judo and 1:3 in BJJ, while in standing position this relation was 1:11 in Judo and 1:2 in BJJ.

Table 2 Technical-Tactical pattern during Judo (n = 16) and BJJ (n = 24) matches and comparisons between modalities.

	BJJ	JUDO				
	Median	P25%	P75%	CV (%)	Median	P25%	P75%	CV (%)	Ux100	p-value	ES	
Overall frequency of actions (n)	
Imobilizations	2.0	0.0	5.2	117.9	1.0	0.0	2.0	107.0	1510	0.240	−0.18	
Guard pass	5.0	3.0	11.0	67.8	0.0	0.0	0.0	400.0	85	<0.001	−0.79	
Chokes	0.5	0.0	1.2	144.1	0.0	0.0	0.0	230.9	1270	0.041	−0.31	
Joints locks	1.0	0.0	2.2	148.1	0.0	0.0	0.0	273.2	1050	0.006	−0.42	
Lever throws	1.5	0.7	3.0	102.9	2.0	1.0	4.0	67.9	1475	0.217	−0.19	
Couple throws	2.0	1.0	4.0	75.0	3.0	1.0	4.2	75.8	1755	0.653	−0.07	
Frequency of effective actions (n)	
Imobilizations	0.0	0.0	1.0	156.9	0.0	0.0	1.0	153.9	1770	0.649	−0.07	
Guard pass	0.5	0.0	1.2	145.8	0.0	0.0	0.0	0.0	960	<0.001	−0.58	
Chokes	0.0	0.0	0.0	199.1	0.0	0.0	0.0	214.9	1880	1.000	−0.02	
Joints locks	0.0	0.0	0.0	328.3	0.0	0.0	0.0	0.0	1520	0.124	−0.29	
Lever throws	0.0	0.0	1.0	134.7	1.0	0.0	1.2	103.2	1580	0.339	−0.15	
Couple throws	1.0	0.0	1.0	144.3	0.0	0.0	1.0	143.8	1305	0.065	−0.28	
Overall values	
Total of actions	15.5	12.7	20.2	52.3	7.0	5.5	9.0	44.6	425	<0.001	−0.62	
Effective action	3.0	1.0	4.2	125.1	1.5	1.0	4.0	81.9	1465	0.206	−0.19	
Efectiveness (%)	19.6	11.9	24.1	66.4	26.7	16.0	44.4	57.5	1195	0.045	−0.30	
Notes.

ES Effect size

P25% and P75% Percentiles

CV Coefficient of variation

Values in bold indicate large effect.

Table 3 Time-motion variables from simulated matches, by modalities (n = 40).

	BJJ (n = 24)	JUDO (n = 16)				
	Median	P25%	P75%	CV (%)	Median	P25%	P75%	CV (%)	Ux100	p-value	ES	
Mean HI time/Block (s)	8.5	6.1	10.9	46.6	2.6	2.0	3.0	26.8	70	<0.001	−0.78	
Mean LI time/Block (s)	17.9	14.7	21.5	43.1	11.2	6.8	13.3	38.1	465	<0.001	−0.61	
Mean pause time/Block (s)	8.0	5.8	10.8	67.9	8.0	7.1	8.8	22.2	1835	0.821	−0.04	
Relative HI time (%)	20.8	13.6	29.0	57.4	5.1	3.5	6.0	52.0	450	<0.001	−0.61	
Relative LI time (%)	71.8	61.4	77.5	19.9	65.1	60.3	69.2	10.4	1295	0.086	−0.26	
Relative pause time (%)	2.6	1.2	5.1	109.0	22.0	19.7	26.6	21.3	20	<0.001	−0.79	
Number of HI blocks (n)	14.5	10.7	17.2	49.0	6.0	4.0	7.0	45.7	500	<0.001	−0.59	
Number of LI blocks (n)	24.5	20.7	26.2	24.4	20.0	15.7	28.5	32.0	1615	0.407	−0.13	
Number of pause blocks (n)	1.5	1.0	4.0	102.0	8.0	7.7	10.5	29.9	205	<0.001	−0.72	
E:P ratio	22:1				3.5:1							
HI:LI ratio	1:3.5				1:12							
Notes.

HI High Intensity

LI Low Intensity

E:P Effort:Pause ratio

HI:LI High:Low intensities ratio

Figure 1 Time-motion analysis description relative to total fight time, considering modalities and positions of actions (Judo n = 16; BJJ n = 24.)

Regression models

Regression models were build based on significant correlations that showed coefficients higher than 0.40. Only those models that presented at least one significant predictor were mantained. In this context, Tables 4 and 5 presented the regression models based on physical fitness variables as predictors. For BJJ (Table 4), the C&FAT seems to be one of the most relevant predictors, being, by itself, capable to explain variation in relative HI Time (31%), mean HI time per blocks (24%), HI:LI ratio (23%) and relative frequency of couple throws (21%). In addition, when C&FAT is combined with horizontal jump, IJCT and medicine ball throws, these tests were capable to explain variation in lever throws frequency (41%), frequency of effective actions (53%) and effectiviness (47%). For Judo athletes (Table 5) there is no proeminent physical variable to predict technical-tactical and time-motion patterns, however, combinations of performance in different tests suggest higher prediction values ranging from 41% to 90%.

Table 4 Regression models for prediction of technical-tactical and time-motion variables by physical fitness tests performance in BJJ athletes (n = 24).

Dependent variable	R2	Constant	Predictors	B	β	t	p	
Relative HI time (%)	0.31	5.89	C&FAT	0.34	0.63	3.07	0.006	
Mean HI time/Block (s)	0.24	14.7	C&FAT	0.08	0.49	2.65	0.01	
HI:LI ratio	0.23	1.01	C&FAT	0.007	0.48	2.6	0.01	
Couple throws (%)	0.21	31.6	C&FAT	0.23	0.46	2.45	0.02	
Lever throws (n)	0.41	6.7	C&FAT	0.03	0.55	2.6	0.01	
HJ	0.02	0.40	1.9	0.07	
IJCT	0.05	0.44	2.3	0.03	
MEDBALL	0.67	0.54	2.7	0.01	
Effective actions (n)	0.53	28.8	C&FAT	0.10	0.47	2.5	0.02	
HJ	0.08	0.40	2.1	0.04	
IJCT	0.25	0.62	3.6	0.002	
MEDBALL	2.7	0.62	3.5	0.002	
Efectiviness (%)	0.47	69.2	C&FAT	0.27	0.67	3.1	0.02	
HJ	0.25	0.54	2.5	0.02	
IJCT	0.44	0.67	3.1	0.02	
MEDBALL	7.7	0.67	3.1	0.02	
Notes.

HI High Intensity

LI Low Intensity

E:P Effort:Pause

VJ Vertical Jump

HJ Horizontal Jump

DJCT Dynamic Judogi Chin-up Test

IJCT Isometric Judogi Chin-up Test

MEDBALL Medicine ball throws

HR Heart Rate

C&FAT Cunningham and Faulkner Anaerobic Test

Table 5 Regression models for prediction of technical-tactical and time-motion variables by physical fitness tests performance in Judo athletes (n = 16).

Dependent variable	R2	Constant	Predictors	B	β	t	p	
Relative HI time (%)	0.90	132.9	FLEXIBILITY	1.01	1.12	5.1	0.001	
DJCT	0.99	0.72	2.6	0.03	
IJCT	0.42	1.48	5.7	0.001	
MEDBALL	5.7	0.66	4.2	0.003	
SIT-UPS	0.71	1.33	6.7	0.001	
HRmax	0.17	0.55	4.6	0.002	
E:P ratio	0.49	3.9	DJCT	0.36	1.03	2.9	0.01	
IJCT	0.09	1.22	3.5	0.004	
HI:LI ratio	0.53	8.4	HRmax	0.04	0.62	3.2	0.006	
C&FAT	0.005	0.33	1.7	0.09	
Couple throws (%)	0.54	257	HRmax	1.2	0.58	3.1	0.008	
			C&FAT	0.19	0.4	2.1	0.05	
Couple throws (n)	0.41	1.07	HJ	0.007	0.63	2.7	0.01	
			DJCT	0.04	0.49	2.1	0.04	
Effective actions (n)	0.68	24.5	FLEXIBILITY	0.15	0.40	2.2	0.04	
VJ	0.11	0.55	3.0	0.04	
VO2max	0.18	0.34	1.8	0.01	
Effectiveness (%)	0.55	10.2	FLEXIBILITY	0.17	0.44	2.4	0.03	
VO2max	0.20	0.58	3.1	0.008	
Notes.

HI High Intensity

LI Low Intensity

E:P Effort:Pause

VJ Vertical Jump

HJ Horizontal Jump

DJCT Dynamic Judogi Chin-up Test

IJCT Isometric Judogi Chin-up Test

MEDBALL Medicine ball throws

HR Heart Rate

C&FAT Cunningham and Faulkner Anaerobic Test

Discussion

The present study aimed to investigate relations between physical fitness variables with technical-tactical and time-motion patterns in Judo and BJJ. The main findings indicate that physical predictors for fight performance are different between Judo and BJJ; and that regression models based on physical fitness variables can explain from 21% to 53% of the variations in technical-tactical and time-motion parameters in BJJ and from 41% and 90% in Judo.

Technical-tactical analysis showed that judokas had higher frequency of lever throws, which is in agreement with previous research that found higher preference for this type of projections in London Olympic Games (Sterkowicz, Sacripanti & Sterkowicz-Przybycień, 2013) and by international level judokas in different competitions (Sterkowicz-Przybycień, Miarka & Fukuda, 2017). The BJJ athletes showed higher frequency of actions applied than judokas, however, it did not resulted in higher effectiveness. In fact, European judokas with lower frequency of actions per fight showed higher effectiveness (24%) than Portugueses and Olympic athletes (Sterkowicz, Sacripanti & Sterkowicz-Przybycień, 2013). In BJJ matches 15 (P25%–75%= 12–20) offensive actions were applied, from those, three (P25%–75%= 1–4) were effective, which is similar to the 8 ± 4 to 14 ± 5 offensive actions and 3 ±1 to 7 ± 2 effective actions previously reported (Andreato et al., 2015b). In Judo matches, the frequency of 7 (P25%–75%= 5–9) offensive actions is close to the Olympic pattern of 6 (P25%–75%= 3–10) actions (Sterkowicz-Przybycień, Miarka & Fukuda, 2017). Our data confirms that judokas seems to apply lever projections while BJJ athletes focus in inmobilizations, guard passages and joint locks. Higher energetic and biomechanical efficiency of lever throws could explain this Judo pattern, which is in agreement to the main Judo principle of “maximal efficiency with minimum effort” (Sterkowicz, Sacripanti & Sterkowicz-Przybycień, 2013). For BJJ, the percentual time dedicated to couple throws sweeps (Fig. 1) is in agreement with previous findings (Del Vecchio et al., 2007; Andreato et al., 2015b).

Previous time-motion analysis on 33 matches from the 2005 BJJ World Cup indicated that effort lasted 170 s(Ground work= 146 ±119 s; Standing: 25 ± 17 s) and pauses lasted 13 ± 6 s, resulting in an E:P ratio of 10:1. Based on 22 regional level matches, Andreato et al. (2013) reported that combats presented 2 ± 1 effort blocks lasting 126 ± 79 s, in a total of 264 ± 103 s of effort; and 2 ± 1of pause blocks of 20 ± 14 s, in a total of 33 ± 25 s of pause, which indicate an E:P ratio of 6:1. From the proposed protocol, combats presented 3 (P25%–75%= 1–8) effort blocks of 165s (P25%–75%= 68–570), in a total of 585s (P25%–75%= 560–601) of effort; and 1.5 (P25%–75%= 1–4) pause blocks of 8 s(P25%–75%= 5–10), in a total of 15 s(P25%–75%= 7–31) of pause, which indicate an E:P ratio of 22 (P25%–75%= 5–46):1. This discrepancy in E:P ratios can be explained by the high variability, probably caused by the absence of pause blocks in five of the measured combats. In addition, these differences remain when 10-min combats are considered, in which a E:P values evidenced was 9:1, the crucial point seems to be the durations of pauses that lasted 78 ± 32 s (13% of total fight) while pauses here presented are similar to previous reported (Andreato et al., 2013). Finally, differences can be due official and simulated combats characteristics.

Specifically considering intensity of the efforts, BJJ combats are composed by 8 ± 3 blocks of 4 ± 4 s of high intensity activity, in a total of 24 ± 14 s; by 10 ± 3 blocks of 25  ± 9 s of low intensity activity, in a total of 224 ± 94 s, which indicate an HI:LI ratio of 1:6 (Andreato et al., 2013). When four successive combats are considered, the HI:LI ratio varies from 1:8 to 1:13, where duration of high intensity actions remains between 2 and 4 s, while low intensity efforts lasted between 27 and 30 s (Andreato et al., 2015b). Results here presented showed a HI:LI ratio of 1:3.5 (P25%–75%= 2–6) that cand be explained by the higher duration of high intensity actions [8 s(P25%–75%= 6–10)], since low intensity actions lasted 18 s(P25%–75%= 15–21), similar than previous studies (Andreato et al., 2013).

Regarding temporal structure of Judo combats, Marcon et al. (2010) analyzed three successive black-belt fights and found that the average duration of technical actions lasted from 1 to 1.4 s, with combat time (preparation, footprint, takedowns and groundwork) ranged from 34 to 38 s, while the mean pause time was 6 to 7 s, with no differences between fights. Data of the present study indicate similar values for pause duration per block [8s (P25%–75%= 7–9) and duration of technical actions [2.6s (P25%–75%= 2–3)], with an E:P ratio of 3.5 (P25%–75%= 3–7):1 and a HI:LI ratio of 1:12 (P25%–75%= 10–18), which points to similarity in the temporal structure of the simulated combats.

To the best of the authors’ knowledge, this is the first study to propose regression models for the prediction of technical-tactical and time-motion pattern in the specialties of grappling in combat sports based on physical fitness variables. Recently, a prediction model for competitive success (score in competitions according to the Croatian federation) of Judo cadet athletes was proposed from physical tests (Kuvačić, Krstulović & Caput, 2017). The findings indicated that maximum strength explained approximately 60% of the variation in performance in the heavier categories for both sexes, which may be associated with less dynamics and greater need for control of the opponent in these categories, while speed and specific endurance were associated with performance in the lighter categories in the male and female, respectively. Previous investigations indicate that, in general, higher aerobic levels (72 ± 2.2 ml/kg/min) has a positive effect on performance in anaerobic exercises in judokas (Franchini et al., 1999). The results indicate a better performance for those with higher maximum oxygen consumption, suggesting that aerobic fitness is, therefore, important for performing total work in high-intensity intermittent actions, such as Judo and BJJ fights. In the same direction, significant correlations were found between the number of projections in the Special Judo Fitness Test and the velocity associated to anaerobic threshold (r = 0.60; p < 0.01) and vVO2max (r = 0.60; p < 0.01) of Judo athletes, as well as an inverse relationship between aerobic capacity and maximal lactate concentration obtained after simulated combat.

Although it is not possible to suggest, from these studies, a relationship between aerobic fitness and technical-tactical aspects with competitive performance (Franchini et al., 1999; Detanico et al., 2012), findings of the present study support those inferences. In this sense, the aerobic component (HRmax and VO2max) seems to significantly predict the frequency of couple throws (16%), chokes (19%) and joint locks (16%) in BJJ and joint locks (25%), effective actions (24%) and effectiveness (35%) in Judo. In addition, when the aerobic and anaerobic components (C&FAT) are combined, the prediction reaches 37% in time at high intensity per block, 53% in the HI:LI ratio and 54% in the percentage of a couple of throws applied in Judo fights. Actually, the demand on the aerobic component presents higher values in the pause periods when compared to the periods of effort, besides being predominant from the first minute of the fight in Judo (Julio et al., 2017). Thus, it reinforces the relevance of aerobic sources in high-intensity intermittent actions, possibly because it is positively associated with the greater capacity to recover between efforts and to the maintainance of the high intensity of the motor actions, even if they have an anaerobic characteristic, which can be mainly due to the phosphocreatine resynthesis rate (Campos et al., 2012; Julio et al., 2017).

However, due to the lactate and glucose concentrations and temporality of Judo and BJJ combats, it is suggested that there is an important request of the glycolytic pathway (Andreato et al., 2015a; Franchini et al., 2016). In addition, because of the very short duration of the high-intensity actions, it is inferred that there is a great demand of the phosphatic system, even with aerobic pathway predominance (Julio et al., 2017). In the present study, the proposed anaerobic fitness test seems to significantly predict time-motion (between 15 and 24%) and technical-tactical variables, as percentage of guard passages (31%) and frequency of joint locks (16%) in BJJ. The role of the C&FAT on Judo was lower, with significant prediction for guard passages and chokes (27% and 22%, respectively). It is also worth mentioning that anaerobic conditioning, although not predominant, may be considered essential for the determinant actions for competitive success (Del Vecchio et al., 2007), since Judo and BJJ matches exhibit high glycolytic activity, inferred by lactate concentrations close to 12 mmol/L after fight (Branco et al., 2013).

When studying Judo athletes, Detanico et al. (2012) observed significant correlations of Special Judo Fitness Test indices with lower limbs power, measured by countermovement jumps (r = 0.74, p < 0.01). Franchini et al. (2015) compared the effects of eight weeks of linear and undulating periodization in Judo athletsfound improvements in some physical fitness variables involving judogi footprint endurance, dynamic strength in basic exercises and performance in a Judo specific test. However, no technical-tactical changes were observed for the number of feints, attacks and attack directions in three simulated fights. It is suggested that the tactical parameters chosen in that study might not be sensitive enough find changes that could have been evidenced through other indicators, such as effective attacks and combat efficiency. This was considered in the present investigation and the findings indicate that, in BJJ fights, performance in upper limb strength and power tests, such as DJCT, IJCT and medicine ball throws, might explain the variations in the frequency of couple throws (31%), chokes (14%), joint locks (24%), effective actions (44%) and effectiveness (27%). For Judo athletes, upper and lower limb power explain 68% of total time at high intensity, while flexibility and abdominal endurance predicts number of attempts, effective actions and effectiveness at 46, 60, and 47%, respectively. In the opposite direction, after applying 14 general and five specific tests with 85 Croatian Karate athletes, Blazević et al. (2006) reported that the best predictors of technical (subjective evaluation of eight techniques by four coaches) and combat (inferred by the competitions results) efficiency were agility (measured with triangle displacements that resemble those of combat) and specific speed, as measured by tests with blocking techniques (gedan barai) and circular kicks (mawashi geri). The authors are still emphatic in indicating that the general tests studied, which involved vertical jumps, horizontal jumps and medicine ball throws, are “unnecessary” and can be “time wasters” for the evaluation of these athletes.

On the other hand, considering grappling modalities, Cvetković et al. (2005) studied the correlation of physical performance with five Wrestling projection techniques. According to the results it is possible to explain from 36 to 41% of the total variance of each of three techniques (hip headlock throw, take-down and throw) from medicine ball throw, 20-m run, and push-ups. Therefore, it is possible to infer that performance in general and specific physical tests can considerably predict performance in technical-tactical and time-motion aspects in grappling combat sports. Finally, the combination of metabolic and neuromuscular fitness variables increases the predictive capacity of the regression models, explaining from 30% to 90% of the time-motion variables associated with high-intensity actions and 47 and 55% in the effectiveness index in BJJ and Judo matches, respectively. Therefore, we evidenced that, even if metabolic and neuromuscular variables explain the performance of different technical-tactical and time-motion actions from differently manners, their combination is essential for competitive performance in these modalities.

From our study design, some limitations need to be highlighted. Firstly, Judo matches duration has been changed from 5 to 4 min; however, it is our opinion that it did not make our finding less relevant. Indeed combat sports rules changes frequently and our data can be considered as a baseline for further research aiming to investigate the effects of the new time structure in Judo fighting pattern. Secondly we focused on adult male subjects and similar investigations are suggested in other aging categories and in female athletes. Third, the convenience method of sample selection should be considered as a limitation. Notwithstanding we believe that the characteristic of the variables measured (performance and technical -tactical parameters) minimized this limitation. Finally, we proposed a design based on simulated matches, for that, the transfer of this knowledge for the real competitive environment requires some caution, as such, we suggest for further studies to replicate this design in official combats.

In summary, our findings suggest that the proposed mobile application was a viable, practical and valid tool for notational analysis and analysis found that correlations between physical fitness and fight performance were not the same for Judo and BJJ. This information reingforce the need for further search to gather knowledge about the specificity of combat sports training.

Practical Applications

Metodological choices for this study design aimed to raise external validity as we believe it could induce higher practical applications. Based on the strength of our findings, we believe that it was a successful choice that can contribute to technical-tactical and physical combat sports training. Firstly, notational analysis are encouraged for coaches and trainers for better understand fight pattern of their own athletes and for opponents. Our results indicate that it can be done with mobile applications, which is an agreement to actual trends and raises logistics, quickness and mobility without validity problems. Secondly, we successful adapted a biomechanical model of analysis for BJJ based on Judo previous protocol. We suggest that it is especially positive for multi-modalities athletes, for those who underwent in mixed combat sports as Mixed Martial Arts and for other modalities as Olympic Wrestling. Finally, we showed that simple, practical and inexpensive tests were capable to predict Judo and BJJ performance. In addition, we presented where and how much each test contributes, considering both technical-tactical and time-motion parameters.

We also suggest that our results could be used as a guide for coaches, trainers and athletes. For an objective example, based on these tests battery and notational analysis, a BJJ athlete with low values of relative high-intensity time would benefit improving and monitoring the C&FAT performance. On the other hand, a Judo athlete with the same deficiency would need to monitor in order to identify if his problems involved flexibility, isometric or dynamic footprint, upper limbs power, core endurance or aerobic fitness.

Conclusion

Prediction models for combat performance here presented indicate that, in Judo, a combination of anaerobic, aerobic and neuromuscular fitness contribute to explain time-motion variables related to high intensity and technical-tactical patterns as frequency, success and effectiveness of actions. On the other hand, in BJJ, only anaerobic fitness seems to be capable to predict time-motion variables related to high intensity, while technical-tactical patterns seems to be affected by both metabolic and neuromuscular parameters.

Supplemental Information

Supplemental Information 1 Raw data

Click here for additional data file.

Additional Information and Declarations

Competing Interests

Author Contributions

Human Ethics

Data Availability

The authors declare there are no competing interests.

Victor S. Coswig conceived and designed the experiments, performed the experiments, analyzed the data, contributed reagents/materials/analysis tools, prepared figures and/or tables, authored or reviewed drafts of the paper, approved the final draft.

Paulo Gentil and Fabrício B. Del Vecchio conceived and designed the experiments, analyzed the data, contributed reagents/materials/analysis tools, authored or reviewed drafts of the paper, approved the final draft.

João C.A. Bueno, Bruno Follmer and Vitor A. Marques performed the experiments, contributed reagents/materials/analysis tools, authored or reviewed drafts of the paper, approved the final draft.

The following information was supplied relating to ethical approvals (i.e., approving body and any reference numbers):

The study was approved by the Federal University of Goiás research ethics committee (no 1.679.225/2016) and was performed in accordance with the Declaration of Helsinki ethical standards.

The following information was supplied regarding data availability:

The raw data are provided in a Supplemental File.

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
