# Peer review of "Physical fitness predicts technical-tactical and time-motion profile in simulated Judo and Brazilian Jiu-Jitsu matches"

_PeerJ, doi:10.7717/peerj.4851_

## Round 0.1 · original submission · Major Revisions

Dear authors:

Your manuscript was evaluated by expert reviewers.

Please address the reviewer´s concerns in detail.

·

Basic reporting

Review the cites throughout the paper
- After of et al “.,” (i.e. Sacripanti et al., 2008)
- When there are several references separated with “;” (change “,” x “;”)
- Change: “and” X “&”
- When the author is directly cited the year goes between (). For example, line 209, by Sacripanti et al. (Sacripanti 2008) x Sacripanti et al. (2008).

Experimental design

More details should be added to the inclusion criteria of the participants (lines 107 a 110). For example, years of practice, frequency of weekly training and competitive level reached. This to guide the practical implications of the readers.

Validity of the findings

This job that presents interesting contributions towards the field of combat sports, especially for the specialties of grappling.

Additional comments

Good work. just ask them to review the following:

Review the cites throughout the paper
Line 71: Gi, describe it the first time, thinking about readers who are not Martial Arts and combat sports.
Line 81: change “inestigations” x “investigations”
Line 189-191: cite and reference the site of the rules of the International Judo Federation and International Brazilian Jiu-Jitsu Federation.
Line 219 a 233 Time motion analysis. Because the intensity of the efforts is assessed subjectively, it is necessary to indicate who valued it. For example, high-ranking professor of Judo and BBJ or researchers with a high level of specialization in the disciplines, etc.
In Table 2, add the number of athletes: BBJ (n = 24); Judo (n = 20).
Line 347, “in belt and jacket combat sports” change for “in the specialties of grappling in combat sports”
Line 426 about limitations. I see a limitation that should be pointed out in the discussion that corresponds to the selection of the sample.

·

Basic reporting

no comment

Experimental design

must include the reference "sit and reach test"
must incorporate the reference of Power of the lower limbs (horizontal jump)
must incorporate the reference of Upper Limbs Power (Medicine Ball Throw)
must incorporate the Upper Limbs Endurance reference (flexion test)
must incorporate the abdominal resistance reference (sitting test)
must incorporate the reference for the calculation and classification of the "effect size"
must review the citations and references format

Validity of the findings

no comment

Additional comments

Should review aspects of the format in general, mainly citations and references.
All tests must be cited and referenced.
Rhe "effect size" must be cited and referenced.

·

Basic reporting

1. Review line 200. The word analysis is repeated.

2.Reading lines 86-88, where the phrase "in this context, the present investigation aimed to study the relationship between physical fitness variables and competitive success, through technical-tactical analysis in judo and bjj athletes"appears. It seems to me that relate the variables related to physical fitness with sports success was not the objective of the study. Given that, this relationship is analyzed with simulated combat and not with a tactical technical analysis in real competition conditions, and the competition is much more complex than a simulated condition in a laboratory. What I recommend is to leave that sentence like as:"In this context, the present investigation aimed at studying the relationship between physical fitness variables and technical-tactical analysis in judo and bjj athletes”.

3.Regarding the revision of the data table (Excel) there is a difference with the text. In the Excel file appears n = 24 BJJ athletes and n = 16 Judo athletes, while in the text appears n = 24 BJJ athletes and n = 20 Judo athletes, check and correct that.

Experimental design

1. In the subjects section, in the paragraph of lines 103 to 106, I suggested to provide more information about the subjects, such as frequency of training, weekly hours of training and years of experience. This would be useful for the practical application of the study. In this way, future readers will be able to understand in a better way the level of the selected subjects, and understand if they were competitive or recreational athletes.

Validity of the findings

no comment

Additional comments

no comment

---

## Round 0.2 · accepted · Accept

I am pleased to inform you of the official acceptance of your manuscript for publication in PeerJ.

Thank you very much for the opportunity to review your manuscript and congratulations.

·

Basic reporting

No comments.

Experimental design

No comments.

Validity of the findings

No comments.

Additional comments

Good job.

·

Basic reporting

All modifications were made.

Experimental design

All modifications were made.

Validity of the findings

All modifications were made.

Additional comments

All modifications were made. The work is of high quality and a contribution to the knowledge of combat sports training